# Changes in Cortisol and in Oxidative/Nitrosative Stress Indicators after ADHD Treatment

**DOI:** 10.3390/antiox13010092

**Published:** 2024-01-12

**Authors:** Laura Garre-Morata, Tomás de Haro, Raquel González Villén, María Luisa Fernández-López, Germaine Escames, Antonio Molina-Carballo, Darío Acuña-Castroviejo

**Affiliations:** 1UGC of Laboratorios Clínicos, Hospital Universitario Clínico San Cecilio, 18016 Granada, Spain; laura.garre.sspa@juntadeandalucia.es (L.G.-M.); tomas.haro.sspa@juntadeandalucia.es (T.d.H.); 2Ibs.Granada, 18016 Granada, Spain; mluisafl86@gmail.com (M.L.F.-L.); gescames@ugr.es (G.E.); 3UGC of Pediatrics, San Cecilio University Hospital, 18016 Granada, Spain; rakel87@correo.ugr.es; 4Ibs.CIBERfes, Centro de Investigación Biomédica en Red Fragilidad y Envejecimiento Saludable, 18016 Granada, Spain; 5Centro de Investigación Biomédica, Departamento de Fisiología, Facultad de Medicina, Parque Tecnológico de Ciencias de la Salud, Universidad de Granada, 18016 Granada, Spain; 6Department of Pediatrics, Medicine Faculty, University of Granada, 18071 Granada, Spain

**Keywords:** attention deficit hyperactivity disorder (ADHD), oxidative stress, inflammation, klotho, cortisol, cortisol awakening response (CAR), methylphenidate

## Abstract

Although ADHD is one of the most prevalent diseases during childhood, we still do not know its precise origin; oxidative/nitrosative stress and the hypothalamic–pituitary–adrenal axis are suggested contributors. Methylphenidate, among others, is the main drug used in ADHD patients, but its effects on relevant markers and structures remain unclear. This study, involving 59 patients diagnosed with ADHD according to DSM-5 criteria, aimed to assess changes in cortisol levels (using cortisol awakening response, CAR) and oxidative/nitrosative status with the treatment. Blood samples before and 3 months after treatment with methylphenidate were used to measure oxidative and inflammatory markers, as well as the endogenous antioxidant activity, while saliva samples tracked cortisol awakening response (CAR). The results showed a treatment-related improvement in the redox profile, with the reduction in advanced oxidation protein products (AOPP), lipid peroxidation (LPO), and nitrite plus nitrate (NOx) levels, and the increase in the enzymatic activities of glutathione reductase (GRd) and catalase (CAT). Moreover, the area under the curve (AUC) of CAR increased significantly, indicating increased reactivity of the HPA axis. These results support, for the first time, the involvement of the endogenous antioxidant system in the pathophysiology of ADHD.

## 1. Introduction

Attention Deficit Hyperactivity Disorder (ADHD) is a neurodevelopmental disease, the main symptoms of which are inattention, hyperactivity, and impulsivity [1]. It is one of the most common mental disorders affecting children, with a prevalence of around 5–8% of the children population [2]. During school years, ADHD is diagnosed more frequently in males (four boys/one girl) with a predominance of hyperactive–impulsive presentation, while in adolescence and young adulthood the ratio changes to one man/two women with predominance of inattentive presentation. It is estimated that two thirds of affected children continue with symptomatology into adulthood, which translates into a prevalence setting at 2–3% of the adult population [3,4]. Another important feature of ADHD is its high comorbidity: about 85% of patients have other associated diseases, making it a disease with an important impact on the public health system [5].

The etiology of ADHD is not well known, although it appears to be a multifactorial disease where genetic inheritance plus environmental interaction is suggested. Many studies justify the hereditary condition of ADHD, estimating a 76% heritability [6]. Increasing evidence suggests that several biological stress-related systems, including the hypothalamic–pituitary–adrenal (HPA) axis, play a key role in this disorder [7]. HPA participates in the regulation of neurotransmitters and stress response, and ADHD patients present a low reactivity of the HPA axis when they are exposed to stressful situations [8]. Activation of the HPA axis involves the release of the glucocorticoid cortisol that follows a diurnal cycle peaking at early morning and progressively declining until midnight [9].

Independent of this circadian variation, there is also a marked increase in cortisol secretion after awakening, known as “Cortisol Awakening Response” (CAR), that consists of a specific rise, around 50–75%, in cortisol levels and occurs within 30–45 min after awakening in the morning and is the most appropriate measure of HPA activation [10,11]. Variations from the usual CAR pattern are assumed to mark maladaptive neuroendocrine processes [12], and reduced cortisol awakening responses have been reported in children with ADHD compared with healthy control children [13].

Additionally, the other stress-related system associated to ADHD comprises oxidative stress and neuroinflammation. Oxidative stress is characterized by a discrepancy between the production of reactive oxygen species (ROS) and the cell’s ability to counteract these effects through its antioxidant defense [14]. Oxidative stress leads to protein and lipid oxidation and damage to DNA structure, which, together with catecholaminergic dysregulation and other genetic and environmental factors, results in a context of neurotoxicity and inflammation [15,16,17,18,19,20]. Recent studies report high levels of oxidative damage with a decreased activity of some antioxidant enzymes in ADHD patients, supporting the theory that oxidative stress could be considered as another pathophysiological factor in this disease [16,21].

Regarding ADHD therapies, an appropriate strategy consists of a multimodal treatment combining psychological and behavioral therapy with pharmacological measures, where the main protagonists are methylphenidate and lisdextroamfetamine (stimulants) and atomoxetine and guanfacine (non-stimulant drugs) [22,23]. However, how these therapies affect the HPA axis and the redox system is yet unknown. For this reason, the main purpose of the present study was to measure the levels of cortisol and different markers of oxidative stress to determine whether clinically significant differences occurred after the usual treatment of ADHD. Additionally, we also evaluated whether, within our patients, there were differences between the prepuberal and puberal groups.

## 2. Materials and Methods

### 2.1. Patients

The study included 59 participants classified into prepuberal and puberal children according to the Tanner scale [24]. The inclusion criteria in both groups were diagnosis of ADHD, normal intellectual ability (Kauffman Brief Intelligence Test), absence of other pathologies (except for the comorbidities typical of these patients) and absence of pharmacological treatment. Exclusion criteria were absence of informed consent, intellectual disabilities, presence of other chronic diseases and patients on pharmacological treatment. The protocol was approved by the Hospital Universitario Clínico San Cecilio’s Ethical Committee (no. 0250-N-20).

### 2.2. Research Design

The study was conducted between September 2020 and May 2023. It is a prospective, quasi-experimental, longitudinal follow-up study based on daily clinical practice. All subjects provided written informed consent and were submitted to a structured clinical interview and a blood and saliva test before starting the usual prescribed treatment and following a 3-month treatment. VARS (Vanderbilt) scales were provided to parents and teachers to be completed. Diagnosis was assigned based on the classification criteria of the Diagnostic and Statistical Manual of Mental Health, fifth edition (DSM-5) [1] and, in addition, the patients were subclassified into different groups based on their presentation of ADHD: Inattentive predominant presentation, hyperactive/impulsive predominant presentation or combined presentation. The main clinical and demographic characteristics are shown in Table 1. The study included 59 participants divided into two groups. The prepuberal children group constituted 34 participants: 15 boys and 8 girls, with a mean age of 7.74 ± 1.29 years. A total of 14 children had ADHD inattentive presentation, another 14 children had combined presentation and the rest of children (6) had hyperactive/impulsive ADHD presentation. The group of puberal children consisted of 25 participants: 14 boys and 11 girls, with an average age of 12.7 ± 1.03 years. Overall, 15 children had inattentive presentation ADHD, another 7 had combined presentation, and the remaining children (3) had hyperactive/impulsive ADHD. All participants received methylphenidate as ADHD treatment.

### 2.3. Blood Samples

Blood samples were obtained before treatment and following a 3-month treatment. These samples were collected from the antecubital vein in the early morning in vacutainer tubes with EDTA-K3. After spinning at 3500 rpm for 15 min at 4 °C, plasma and red blood cells were separated, and erythrocytes were washed twice with cold 0.9% NaCl. These were aliquoted and frozen at −80 °C until analysis. On the day of the experiment, washed erythrocytes were thawed and hemolyzed in phosphate buffer, deproteinized with ice-cold 10% trichloroacetic acid, and centrifuged at 20,000× *g* for 15 min. The supernatants obtained were then employed for analysis.

### 2.4. Salivary Samples

To assess the CAR, participants were asked to take three saliva samples at home: immediately at awakening, 30 and 60 min thereafter. Salivary samples were collected using Salivette© devices (Sarstedt, Nümbrecht, Germany). Family members recorded in writing the time at which each sample was taken. Subjects were instructed not to eat and not to brush their teeth at least 30 min before completing saliva sampling to avoid contamination of saliva. Participants were given a protocol with detailed sampling instructions. On the day of collection, the patients were also scheduled at the hospital for blood analysis where they handed in their saliva samples. Samples were centrifuged at 3000× *g* for 10 min and then frozen at −80 °C until the assays were performed. As with blood samples, saliva samples were also collected at the beginning of the study (before starting treatment) and three months later.

### 2.5. Measurement of Advanced Oxidation Protein Products (AOPP)

AOPP was quantified spectrophotometrically on a microplate reader (PowerWaveX; Bio-Tek Instruments, Inc., Winoosky, VT, USA). The measurement is based on the detection of dityrosine (a specific marker of protein oxidation) which, upon reaction with potassium iodide and acetic acid, formed a component that absorbs at 340 nm. For this purpose, a calibration curve was performed using serial dilutions of chloramine T. Next, in each of the sample wells of the microplate, 200 μL of sample diluted, PBS and acetic acid were added, and absorbance was measured against a blank. The AOPP levels was expressed as nmol/mL of chloramine-T equivalents.

### 2.6. Measurement of Products of Lipid Peroxidation (LPO)

LPO levels were measured in plasma by a commercial kit (KB03002, Bioquochem S.L., Oviedo, Spain) which measured malondialdehydes (MDA) and 4-hydroxynonenal (HNE) concentrations as indicators of lipid peroxidation. The reaction produced a chromophore with a maximum absorbance at 586 nm, which was read by a plate reader. LPO concentrations were expressed in nmol/mL [25].

### 2.7. Nitrite + Nitrate (NOx) Measurement

Since nitric oxide (NO) is a very unstable molecule, direct measurement of its content is difficult, so an indirect method of greater stability, such as nitrite and nitrate determination (NOx), is often used. NOx concentration was determined using the Griess reaction, which produces a compound spectrophotometrically detected at 550 nm [26]. For this purpose, plasma samples were treated with 6% sulfosalicylic acid to precipitate proteins. Then, they were centrifuged and the supernatant obtained was combined with 1.25% NaOH and a mixture consisting of nitrate reductase (NRd), glucose-6-phosphate dehydrogenase (G6PDH), glucose-6-phosphate (G6P) and NADPH, in order to reduce nitrates to nitrites. It was left to incubate at room temperature for one hour, and finally, 100 µL of Griess reagent (1:1 mixture of 0.1% Naphthyl-ethylenediamine and 1% sulfanilamide in 5% H3PO4) was added to each sample well. Absorbance was measured at 550 nm in a plate spectrophotometer (PowerWaveX; Bio-Tek Instruments, Inc., Winoosky, VT, USA). Their values were reported in nmol/mL.

### 2.8. Measurement of Glutathione Peroxidase (GPx), Glutathione Reductase (GRd) and Catalase (CAT)

GPx, GRd and CAT constitute the first line of antioxidant defense of our body. Their activities were determined in the erythrocyte fraction. GPx activity was spectrophotometrically measured in a 96-well plate (PowerWaveX; Bio-Tek Instruments, Inc., Winoosky, VT, USA). The method consisted of an indirect measurement of NADPH consumption. GRd activity was also measured spectrophotometrically, using a commercial kit (703202; Cayman chemical, Ann Arbor, MI, USA). Oxidation of NADPH to NADP+ resulted in a decrease in absorbance at 340 nm, which was directly proportional to GRd activity.

CAT activity was measured by the rate of H_2_O_2_ decomposition, according to the method of Aebi [27]. All enzyme activity was expressed as µmol/min/g Hb. Hemoglobin concentration was spectrophotometrically determined by Drabkin’s method [28].

### 2.9. Measurement of α-Klotho

α-Klotho plasma levels were measured by solid-phase sandwich ELISA (Bionova Científica S.L, Madrid, Spain), which relies on two specific antigen–antibody reactions and semiquantitative measurement by spectrophotometry. This assay has a 96-well plate format and each well contains a soluble monoclonal anti-human (IgG) α-Klotho monoclonal antibody. To carry out the reaction, the samples and the standard are first added to the wells and incubated at 23 °C for 60 min for the first reaction to occur. Next, a wash is performed to remove the fraction of sample unbound to the mobilized antibody, an HRP-conjugated secondary antibody is added. It is incubated for another 30 min for the second reaction to take place. After washing again, a solution of tetramethyl benzidine (TMB) and sulfuric acid is added to each well, causing a color change that is measured spectrophotometrically at 450 nm. α-Klotho levels were reported in pg/mL.

### 2.10. Salivary Cortisol

Measurement of salivary cortisol is a good surrogate for plasma cortisol because it correlates well with free serum cortisol and is relatively stable even at room temperature. Salivary cortisol was be measured by liquid chromatography tandem-mass spectrometry (LC-MS/MS). This method is considered as the reference technology for measuring salivary cortisol, among other reasons, because of its high sensitivity and specificity, its standardization and because it avoids cross-reactivity problems with other steroids. Measurement of salivary cortisol was performed with commercial kit MassChrom^®^ Cortisol (Chromsystems Instruments & Chemicals GmbH, Gräfelfing, Germany). Saliva samples were prepared following the manufacturer’s instructions and 15 μL of each sample was injected into the instrument at a flow rate of 0.5 mL/min. Quantification was carried out on a Nexera XR LC liquid chromatography instrument (Shimadzu, Kyoto, Japan) coupled with a QTRAP 4500 LC-MS/MS4500 mass spectrometer (AB Sciex, Framinham, MA, USA). Chromatographic separation was achieved by means of a Chromsystems analytical column (Chromsystems Instruments & Chemicals GmbH, Gräfelfing, Germany) using the gradient elution profile. Results were expressed in μg/L.

### 2.11. Data Analysis

All data are expressed as mean ± SEM. Data were analyzed by unpaired one-way analysis of variance (ANOVA) followed by Student’s test to identify significant differences between all variables. Two-way ANOVA was also used to identify the influence of puberal state and treatment on the results. A correlation and regression analysis was also performed between the variables in PreT and PostT groups. Statistical analysis was conducted using GraphPad Prism 8.0 for Windows (GraphPad Software Inc., La Jolla, CA, USA) for all analyses. A *p* value of <0.05 was considered statistically significant.

In the particular case of cortisol, for the assessment of CAR, this was coded as present when the second measurement was at least 50% higher than the baseline [10,29]. In addition, the area under the curve (AUC) was also calculated according to Pruessner‘s method. The mean cortisol concentration at each of the three time points draws a curve representing the CAR. Pruessner et al. designed two formulas to calculate the area under the curve: one with respect to the ground (AUC_G_) and the other with respect to the increment (AUC_I_) [30]. Subjects who were missing any of the three samples were excluded from the analyses.

## 3. Results

### 3.1. Oxidative and Inflammatory Status in the Patients before and after Treatment

The characterization of oxidative/nitrosative status of ADHD patients was determined by analyzing plasma AOPP, LPO and NOx levels. After three months of treatment, we observed a marked decrease in LPO levels (*p* < 0.001), AOPP (*p* < 0.05), and NOx levels (*p* < 0.05) (Figure 1A–C). CAT, and the enzymes of the glutation cycle, GPx and GRd, were intracellularly measured in erythrocytes. The enzymatic activity of GRd and CAT increased after treatment (*p* < 0.05, Figure 1E,F), whereas GPx activity did not change (Figure 1D). Regarding α-Klotho levels, these decreased after treatment, but not significantly (Figure 1G).

### 3.2. Assessment of Oxidative and Inflammatory Status as a Function of Maturational Stage: Prepuberal vs. Puberal Children’s Groups

To explore whether the stage of pubertal development had any influence on the oxidative and inflammatory status elsewhere measured, we categorized ADHD patients into different maturational stages (I–IV) according to the Tanner scale. Patients classified as Stage I formed the prepuberal children group (*n* = 33), while those in Stages II-IV constituted the group of puberal children (*n* = 25). With this classification, we observed a significant decrease in the levels of LPO in prepuberal (*p* < 0.001) and puberal (*p* < 0.05) patients after treatment (Figure 2B), and only in prepuberal patients AOPP decreased with treatment (*p* < 0.05, Figure 2A). NOx concentrations also decreased in puberal but not in prepuberal subjects after treatment (*p* < 0.05, Figure 2C). Regarding the enzyme activities, we found a significant decrease in GPx in treated puberal subjects (*p* < 0.05, Figure 2D) and an increase in CAT in treated prepuberal patients (*p* < 0.05, Figure 2F). No changes were found in GRd activity in any group (Figure 2E) and neither in α-Klotho levels (Figure 2G).

Correlation analyses performed on the different variables measured before and after treatment showed no significance.

### 3.3. Salivary Cortisol Levels

A total of 354 saliva samples (59 participants × 3 samples × 2 collections) were received in the laboratory. Of all of these, 27 were excluded from analyses due to missing some data points, contamination with blood or blue coloration, which would indicate a very probable contamination due to tooth brushing, among other causes. Mean cortisol levels for all sampling points, in prepuberal and puberal children group, before and after treatment are displayed in Figure 3.

There were no differences in cortisol levels just upon waking up between treated and untreated subjects. At 30 min, however, the levels of cortisol were significantly higher in patients after receiving the treatment compared with those in untreated patients (*p* < 0.05, Figure 3A). No differences were found at 60 min. In prepuberals (Figure 3B), cortisol levels at 30 min, but not at 60 min, were significantly higher than those corresponding to the moment of awakening both in treated and untreated patients (*p* < 0.01). Postpuberal children only showed significant changes in cortisol at 30 min in the treated group (*p* < 0.05, Figure 3C).

On the other hand, considering a positive CAR when the cortisol concentration at 30 min was 50% higher than when waking up [10,29], we observed that in the prepuberal group, the rate of positive CAR before treatment was 53.33%, increasing up to 58% after treatment. In the puberal children group, these differences increased, since the rate of positive CAR before treatment was 48% and 60% after treatment.

The area under the curve (AUC) was also calculated according to Pruessner’s method [30]. Our results showed that AUC_G_ after treatment was higher than AUC_G_ before it (AUC_G_ posT = 33.52 vs. AUC_G_ preT = 23.50; *p* < 0.05). The same happened for AUC_I_ (AUC_I_ posT = 10.98 vs. AUC_I_ preT = 4.43; *p* < 0.05). In addition, AUC_I_ also was significantly higher after treatment in the puberal children group (AUC_I_ posT = 12.33 vs. AUC_I_ preT = 3.35; *p* < 0.05).

Correlation analysis between biochemical variables and CAR was performed in PreT and PosT and in PreT vs. PosT groups, but no correlation was found between them, and thus, no further information could be obtained from this analysis.

## 4. Discussion

Despite the high prevalence and comorbidity of ADHD, we still do not know the exact neurobiological basis of the disorder, although it seems to be related to genetic, environmental and anatomical factors [5,31]. We show here for the first time that the oxidative and inflammatory status of children diagnosed with ADHD, as well as the function of the hypothalamus–pituitary–adrenal axis, improved after pharmacological treatment with the psychostimulant methylphenidate. Thus, our data further support that stress and inflammation, which seem to play a key role in several psychiatric diseases, are involved also in the pathophysiology of ADHD [32,33]. 

Some data point to the presence of high levels of stress and oxidative damage in ADHD patients and a decrease in the activity of various endogenous antioxidant enzymes, although great discrepancy exists in the literature [16,21,34,35,36,37,38]. In fact, a meta-analysis found an association between ADHD and oxidative damage but failed to demonstrate alterations of antioxidant systems [39]. These divergences, together with the fact that oxidative stress may not be a causal agent of the disease but just another consequence of it, highlight the need for further research evaluating biomarker modifications before and after pharmacological treatment of ADHD patients to gain a deeper understanding of the disease [21]. The few existing studies on the effects of psychostimulants on oxidative stress show inconsistent results, and they are based on animal models and measurements in different specimens and brain structures [40]. In rats, it was observed that the effect of methylphenidate varied according to age and type of exposure to the drug, and so, young rats showed a reduction in LPO after acute treatment, whereas chronic treatment increased oxidative damage in young and decreased it in adult rats [41]. Another study showed that the effect of the treatment varied according to brain region; in some areas such as cerebellum and striatum, ROS levels were reduced and antioxidant enzyme activity enhanced, while in the prefrontal cortex, worsening of oxidative stress was observed [42]. Similarly, chronic use of methylphenidate in rats had different effects depending on the clinical context: in the animal model of ADHD simulating therapeutic use (Spontaneously Hypertensive rats, SHR), methylphenidate showed benefits by increasing antioxidant capacity and reducing LPO in the hippocampus and the prefrontal cortex. However, in healthy rats used as control (Wistar Kyoto rats, WKY), simulating non-therapeutic use of the drug, methylphenidate had detrimental consequences in terms of oxidative and nitrosative stress, astrocytic reactivity and antioxidant capacity [43]. Comparable results were observed in another study on rats where methylphenidate treatment improved the proinflammatory profile in the ADHD model while in control conditions it induced a proinflammatory state [44].

Our results show that three months of treatment significantly improved oxidative status in terms of reducing protein oxidative damage (AOPP), lipid oxidative damage (LPO), and reducing inflammation measured by the NOx levels. These results agree with those of other authors who also observed an improvement in oxidative status and mitochondrial membrane potential in human SH-SY5Y neuroblastoma cells [45,46]. Even more, our findings also demonstrate that methylphenidate treatment enhanced the protective action of the endogenous antioxidant system with the increase in CAT and GRd. Overall, our data further support the results in ADHD children and adolescents [40], as well as in patients suffering from depression, where treatment with psychotropic drugs resulted in a normalization of some parameters related to oxidative stress [47].

To explain the changes in oxidative status here reported, several hypotheses are proposed. Some studies suggest that the increase in dopamine induced by methylphenidate could have an antioxidant effect since dopamine can scavenge free radicals [45,48]. Other studies indicate that methylphenidate would have protective properties by sequestering toxins and toxic metabolites of dopamine through the redistribution of the vesicular monoamine transporter type 2 (VMAT2) [49,50,51,52]. Alternatively, the beneficial effect of methylphenidate would be exerted by protecting cells against neurotoxic metabolites [53] or by increasing the activity of SOD and CAT. These mechanisms may underly the amelioration of oxidative stress here reported.

Concerning α-Klotho, although there is no direct link between this protein and ADHD in the literature, α-Klotho is highly expressed in the brain and plays an important neuroprotective role [54], which could in part be mediated by the suppression of proinflammatory cytokines in the central nervous system [55], thus establishing a hypothetical link with the proinflammatory theory of ADHD. In fact, its involvement in other neuropsychiatric diseases such as schizophrenia [56] or bipolar disorder [57] has already been studied, and a higher concentration of α-Klotho has been demonstrated in patients with these diseases compared to healthy controls [58]. This could most likely be due to its ability to cope with oxidative stress and its beneficial effect on cognition [56]. In our study, however, we found no significant differences in α-Klotho levels after three months of treatment.

Regarding the participation of the HPA axis in the pathophysiology of ADHD, several studies suggest that its symptoms may be linked to arousal deficits or to an inability to maintain adequate levels of it [59]. A widely recognized way to assess axis function is the CAR [60]. Although their measurement is relatively straightforward, there is no consensus on which formulas to use or which parameters to measure to accurately assess the CAR pattern [10,61,62,63]. In 2009, Pruessner et al. developed two formulas derived from the trapezoid method to measure cortisol secretion after awakening: AUC_G_ and AUC_I_ [30]. These formulas, although similar, differ in their approach since, while AUC_G_ encompasses the entire area under the curve with respect to the ground and represents the total hormone production from a certain point in time, AUC_I_ includes only the area under the curve with respect to the first measurement and focuses on the changes produced in a given period of time. Our study is the first one in evaluating the effect of ADHD treatment on CAR. Previous studies have only investigated the effect of treatment on basal cortisol levels, and their results are sparse and inconsistent; some of them report an increase in cortisol with methylphenidate and atomoxetine [64], other ones report an initial increase followed by a gradual decrease [65], and others detect differences in dehydroepiandrosterone sulfate but not in cortisol levels [66]. In our results, we considered particularly relevant the lack of significant differences in basal cortisol levels (just at awakening) before and after receiving pharmacological treatment. Cortisol levels 30 min after awakening were, however, significantly higher in the treated group compared with those of the untreated one, thus reflecting a variation in the CAR. Moreover, the levels of cortisol at 30 min after awakening were significantly higher compared to the levels just upon awakening in both groups of patients. Thus, although ADHD may affect the reactivity of the HPA axis, it responds with an increase in cortisol levels even in absence of treatment [10,67]. To determine whether there was a significant change in the CAR after receiving treatment, we carried out different strategies already described and elsewhere used by other authors. Considering that CAR is present when the cortisol concentration at 30 min is 50% higher than that just awakening, our data indicate that CAR was present in 50.9% of patients and, after 3 months of therapy, it increased to 59.25%. Although it is still lower than that observed in the healthy population, it increased by almost 10% with methylphenidate [67]. On the other hand, calculating the area under the curve of cortisol data at each time point (just at awakening, 30 min, and 60 min later) and using the formulas proposed by Pruessner et al. [30], we observed that there was a statistically significant increase in both AUC_G_ and AUC_I_ after treatment. Thus, we can infer that, after treatment, patients diagnosed with ADHD presented a better response to cortisol awakening since there was both a significant increase in total cortisol production during this period (AUC_G_) and an adequate increase in the levels of this hormone (AUC_I_). Given that cortisol is the hormone of the day, related to stress and activity, and methylphenidate is a stimulating drug that increases daytime performance, it seems reasonable to consider that the significant increase in cortisol after treatment reflects its positive clinical effect. All this, together with the positive evolution observed in the oxidative and inflammatory profile, could constitute the biochemical basis that would explain the clinical benefits manifested in our patients, such as the strengthening of concentration or memory and the improvement in academic performance, in social behavior and in the management of impulsive acts.

The maturational development did Indeed exert a significant influence on some parameters such as AOPP, nOx, and GRd. In the puberal children group, AOPP levels (a marker indicative of oxidative stress and inflammation) and nOx levels (a marker of inflammation) were lower than those in the prepuberal group. To our knowledge, there is no other study in the literature related to the impact of maturational development on oxidative status and inflammation in ADHD patients. These findings may point to the hypothesis that during the different maturational stages that conform puberty, the antioxidant defense systems decrease, and this is reflected in a significant increase in AOPP and in NOx. The lack of changes in α-Klotho among pre- and puberal groups here reported suggest that this pathway is not involved in the changes reported in ADHD patients. Nevertheless, further studies should be conducted to confirm this hypothesis and to establish more solid conclusions in relation to the different etiologic factors involved in ADHD and the complex relationship between maturational development and oxidative and inflammatory markers and their clinical implications.

## 5. Conclusions

Our data support an improvement of the redox status, inflammatory profile, as well as the response of the HPA axis after three months of treatment. Because our study was conducted with patients and most of the data reported in the literature related to animal models, we believe that the results here shown represent a more reliable profile of the evolution of the ADHD subjects treated with a first-line drug for this pathology.

## Figures and Tables

**Figure 1 antioxidants-13-00092-f001:**
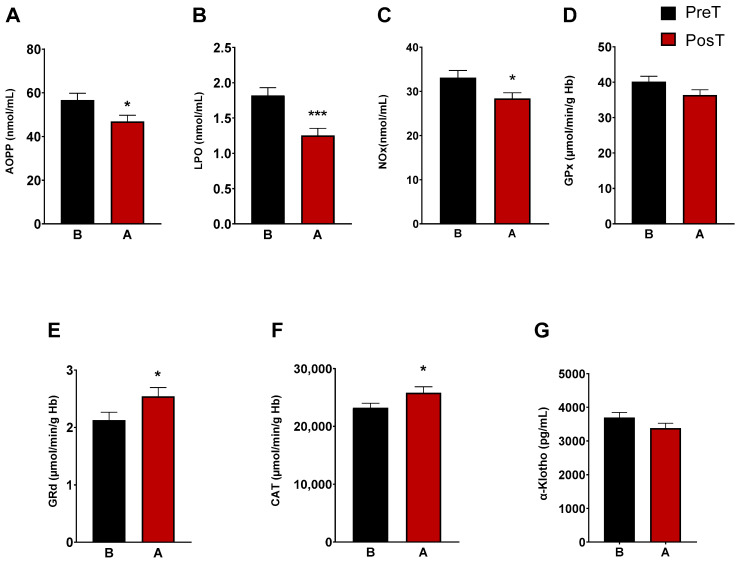
Analysis of plasmatic and erythrocyte oxidative stress parameters in ADHD patients. The following oxidative stress levels are represented: (**A**) advanced oxidation protein product plasma levels (AOPP); (**B**) products of lipid peroxidation (LPO); (**C**) nitrite plus nitrate levels (NOx); (**D**) glutathione peroxidase activity (GPx); (**E**) glutathione reductase activity (GRd); (**F**) catalase activity (CAT); (**G**) α-Klotho. Data are presented as mean ± SEM. PreT, pretreatment; PosT, posttreatment. * *p* < 0.05; and *** *p* < 0.001 vs. PreT.

**Figure 2 antioxidants-13-00092-f002:**
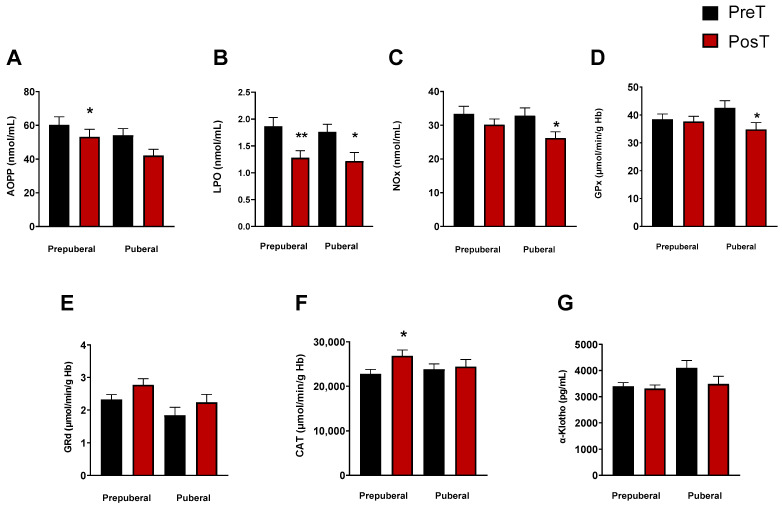
Analysis of plasmatic and erythrocyte oxidative stress parameters in ADHD patients. In each graph, the two columns on the left represent the prepuberal children group and the two columns on the right represent the puberal children group. The following oxidative stress levels are represented: (**A**) advanced oxidation protein products plasma levels (AOPP); (**B**) products of lipid peroxidation (LPO); (**C**) nitrite plus nitrate levels (NOx); (**D**) glutathione peroxidase activity (GPx); (**E**) glutathione reductase activity (GRd); (**F**) catalase activity (CAT); (**G**) α-Klotho. Data are presented as mean ± SEM. PreT, pretreatment; PosT, posttreatment. * *p* < 0.05; ** *p* < 0.01, and vs. PreT.

**Figure 3 antioxidants-13-00092-f003:**
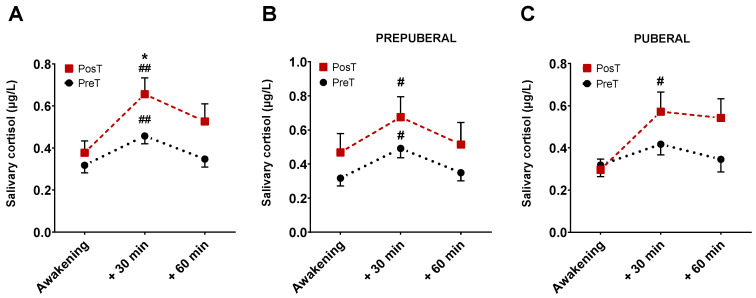
Representation of mean cortisol levels and SEM at three time points, before and after treatment. The graph (**A**) represents cortisol levels without distinguishing between pre-puberty and puberty. The graph (**B**) corresponds to the cortisol levels in the group of prepuberal children and the graph (**C**) with those of the group of puberal children. * *p* < 0.05 vs. 30 min PosT; ^#^
*p* < 0.05, and ^##^
*p* < 0.001 vs. awakening time for each group.

**Table 1 antioxidants-13-00092-t001:** Clinical and demographic characteristics of participants.

Parameters	Prepuberal Group (*n* = 34)	Puberal Group (*n* = 25)
Age (years)	7.74 ± 1.29	12.7 ± 1.03
Gender (female/male)	8 female/26 male	11 female/14 male
ADHD presentation	ADHD combined: 14	ADHD combined: 7
	ADHD inattentive: 14	ADHD inattentive: 15
	ADHD hyperactive-impulsive: 5	ADHD hyperactive-impulsive: 2
	Unclassified: 1	Unclassified: 1
Treatment	Methylphenidate	Methylphenidate

## Data Availability

The data presented in this study are available on request from the corresponding author.

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
