# Peer review of "Changes in Cortisol and in Oxidative/Nitrosative Stress Indicators after ADHD Treatment"

_antioxidants, 2024, doi:10.3390/antiox13010092_

Round 1

Reviewer 1 Report

Comments and Suggestions for Authors

The manuscript by Laura Garre-Morata and colleagues describes the effects of methylphenidate (MPH) treatment in ADHD patients with a focus on markers of oxidative/nitrosative status and the cortisol awakening response. The authors find consistent decreases in lipid peroxidation, advanced oxidation protein products and nitrite + nitrate levels and increases in glutathione reductase and catalase activities after MPH treatment. They also find changes in CAR in patients after MPH treatment. Finally, they assess whether patients in puberal or prepuberal stages exhibit significant differences in their response to MPH.

Overall, this is a novel, well designed and executed study that certainly deserves publication. I only have a couple of comments mainly on the presentation of the results and the statistical analyses.

Comment 1: The authors should perform a correlation and a  logistic regression analysis of all patients and the parameters they measured PreT vs PostT. For example, although the results show consistent changes in AOPP, LPO, NOx, enzymes, as well as CAR, it is not clear if these changes do correlate. Also, correlation or logistic regression analyses would be a suitable way to identify subsets of patients that respond to a greater extent to treatment. For instance, the patents that respond with larger changes in AOPP, LPO, NOx are the same that display a larger CAR effect PosT ?

Comment 2. Results, lines 297-304: The authors should give further details and be more specific on how they calculated AUCs in the Methods Section. 

Minor comment. Discussion, line 333: is there a rat model of ADHD ? please specify .

Author Response

  1. Comment 1: The authors should perform a correlation and a  logistic regression analysis of all patients and the parameters they measured PreT vs PostT. For example, although the results show consistent changes in AOPP, LPO, NOx, enzymes, as well as CAR, it is not clear if these changes do correlate. Also, correlation or logistic regression analyses would be a suitable way to identify subsets of patients that respond to a greater extent to treatment. For instance, the patents that respond with larger changes in AOPP, LPO, NOx are the same that display a larger CAR effect PosT ?

A correlation analysis was done to assess any relationship between the measured variables. No correlation was found between the variables in the groups PreT and PosT, or between PreT vs PosT, and not correlation was detected with CAR. Information of this was added to the Statistics and to the Results, lines 352-354.

  1. Comment 2. Results, lines 297-304: The authors should give further details and be more specific on how they calculated AUCs in the Methods Section. 

Information regarding the calculation of AUC is added to the Material and Methods part (lines 218-224).

  1. Minor comment. Discussion, line 333: is there a rat model of ADHD ? please specify

Yes. The spontaneously hypertensive rat (SHR) is the most well-characterized and commonly used animal model of ADHD [4,23]. This model exhibits neurobiological and behavioral features of this neurodevelopmental condition and it fits the criteria for ADHD diagnosis (https://doi.org/10.3390/antiox12040937). I have added this information to the text (lines 325-328).

Reviewer 2 Report

Comments and Suggestions for Authors

This manuscript evaluates the effect of the drug Methylphenidate, one of the most common drugs used in the treatment of ADHD, on a number of parameters indicative of oxidative stress, of oxidative damage and of antioxidant response, as well as on the Cortisol Awakening Response (CAR). Nitrite and nitrate levels were considered to be an indirect measure of the inflammatory status. The plasma levels of alpha-Klotho were assessed as well. Evaluations were carried out on 34 pre-puberal and 25 puberal children.

Data demonstrate that 3-month administration of the drug ameliorated the oxidative stress and damage, enhanced the antioxidant response, and increased the CAR. Results are not quantitatively large.

The methods chosen by Authors are appropriate and the study was carried out in a correct way. The discussion is sufficiently in-depth.

In my opinion, levels of nitrites and nitrates are not the best way to evaluate the inflammatory status; if possible, levels of the major cytokines should be evaluated as well.

Authors should also introduce the literature suggesting the involvement of alpha-Klotho in ADHD.

The discussion should also consider the results in relation with the known therapeutic effects of Methylphenidate in pre-puberal and puberal children and the differential improvement of oxidative stress markers and of CAR.

Statistical significance of data should be verified after testing for multiple comparisons.

Although the manuscript is easily understandable, there are a number of English problems. Here are the ones that I consider the most striking ones:

line 73: “oxidative stress… results in a potent neuroprotective and inflammatory response” This sentence is ambiguous. Oxidative stress may activate an antioxidant response, but not always, as mentioned in the following line. The inflammatory response and the neuroprotective response may in turn be present or not and, in any case, the former contrasts the latter.

Lines 101-2 “and 3 months later” >>> possible correction as follows: “and following a 3 month-treatment”

Lines 119-20 “Two blood extractions were performed: one before treatment (pre-treatment) and the

other one three months after it (post-treatment).”>>> possible correction as follows: “blood samples were obtained before treatment and following a 3 month-treatment”

line 132 “sampling times” what are they?

Line 284 “after receiving pharmacological treatment.”>>> “between treated and untreated subjects”

Line 289 “pubertal” >>> “post-puberal”

Line 313 “also are involved in the pathophysiology of ADHD” >>> “are involved also in the pathophysiology of ADHD”

Lines 333 and 335 “therapeutic/not therapeutic conditions” is probably not appropriate

Comments on the Quality of English Language

There are a number of sentences that should be modified/corrected

Author Response

  1. Comment: In my opinion, levels of nitrites and nitrates are not the best way to evaluate the inflammatory status; if possible, levels of the major cytokines should be evaluated as well.

  1. Nitrite plus nitrate (NOx) levels depend directly on the activation of iNOS, the inducible nitric oxide synthase. iNOS is one of the main molecules induced under inflammation, together with some proinflammatory cytokines. So, the increase in NOx fit well with inflammation.

  1. Comment: Authors should also introduce the literature suggesting the involvement of alpha-Klotho in ADHD.

  1. Text and bibliography are added to the Discussion (lines 356-362).

  1. Comment: The discussion should also consider the results in relation with the known therapeutic effects of Methylphenidate in pre-puberal and puberal children and the differential improvement of oxidative stress markers and of CAR.

Although some aspects of therapeutical effects of Methylphenidate have been discussed, this study did not focus on the clinical features of the patients, also on biochemical data. Correlations between biochemical and clinical data is, of course, of great interest, and we are preparing other study to answer this question.

  1. Comment:Statistical significance of data should be verified after testing for multiple comparisons.

  1. Multiple comparisons and correlations between the variables have been done (as suggeste also by other reviewer) but not additional significations were found.

Although the manuscript is easily understandable, there are a number of English problems. Here are the ones that I consider the most striking ones:

  • Line 73:“oxidative stress… results in a potent neuroprotective and inflammatory response” This sentence is ambiguous. Oxidative stress may activate an antioxidant response, but not always, as mentioned in the following line. The inflammatory response and the neuroprotective response may in turn be present or not and, in any case, the former contrasts the latter.
  • Lines 101-2 “and 3 months later” >>> possible correction as follows: “and following a 3 month-treatment”
  • Lines 119-20 “Two blood extractions were performed: one before treatment (pre-treatment) and the other one three months after it (post-treatment).”>>> possible correction as follows: “blood samples were obtained before treatment and following a 3 month-treatment”
  • Line 132 “sampling times” what are they?
  • Line 284 “after receiving pharmacological treatment.”>>> “between treated and untreated subjects”
  • Line 289 “pubertal” >>> “post-puberal”
  • Line 313 “also are involved in the pathophysiology of ADHD” >>> “are involved also in the pathophysiology of ADHD”
  • Lines 333 and 335 “therapeutic/not therapeutic conditions” is probably not appropriate

These grammatical changes were done, and the manuscript has been revised for any other grammatical or typographical mistakes.

Round 2

Reviewer 1 Report

Comments and Suggestions for Authors

The authors have addressed all my comments and concerns adequately.